# Association of Low Serum Bilirubin Concentrations and Promoter Variations in the *UGT1A1* and *HMOX1* Genes with Type 2 Diabetes Mellitus in the Czech Population

**DOI:** 10.3390/ijms241310614

**Published:** 2023-06-25

**Authors:** Alena Jirásková, Jan Škrha, Libor Vítek

**Affiliations:** 1Institute of Medical Biochemistry and Laboratory Diagnostics, 1st Faculty of Medicine, Charles University, General University Hospital in Prague, Katerinska 32, 12000 Prague, Czech Republic; alena.jiraskova@immunai.com; 23rd Department of Internal Medicine, 1st Faculty of Medicine, Charles University, General University Hospital in Prague, Katerinska 32, 12000 Prague, Czech Republic; jan.skrha@lf1.cuni.cz; 34th Department of Internal Medicine, 1st Faculty of Medicine, Charles University, General University Hospital in Prague, Katerinska 32, 12000 Prague, Czech Republic

**Keywords:** benign hyperbilirubinemia, bilirubin, Gilbert syndrome, heme oxygenase, HMOX1, type 2 diabetes mellitus, UGT1A1

## Abstract

Bilirubin has potent biological beneficial effects, protecting against atherosclerosis, obesity, and metabolic syndrome. The aim of this study was to assess serum bilirubin concentrations and (TA)_n_ and (GT)_n_ microsatellite variations in the promoter regions of the *UGT1A1* and *HMOX1* genes, respectively, in patients with type 2 diabetes mellitus (T2DM). The study was carried out in 220 patients with T2DM and 231 healthy control subjects, in whom standard biochemical tests were performed. The (TA)_n_ and (GT)_n_ dinucleotide variations were determined by means of fragment (size-based) analysis using an automated capillary DNA sequencer. Compared to controls, both male and female patients with T2DM had lower serum bilirubin concentrations (9.9 vs. 12.9 μmol/L, and 9.0 vs. 10.6 μmol/L, in men and women, respectively, *p* < 0.001). Phenotypic Gilbert syndrome was much less prevalent in T2DM patients, as was the frequency of the (TA)_7/7_
*UGT1A1* genotype in male T2DM patients. (GT)_n_
*HMOX1* genetic variations did not differ between diabetic patients and controls. Our results demonstrate that the manifestation of T2DM is associated with lower serum bilirubin concentrations. Consumption of bilirubin due to increased oxidative stress associated with T2DM seems to be the main explanation, although (TA)_n_ repeat variations in *UGT1A1* partially contribute to this phenomenon.

## 1. Introduction

Bilirubin belongs to the most potent endogenous antioxidants [1], and recently its potential endocrine effects have also been recognized [2,3,4]. Mildly elevated serum bilirubin concentrations protect against both coronary [5] and carotid atherosclerosis [6], as well as obesity and metabolic syndrome [7,8,9]. Negative associations were also reported between serum bilirubin concentrations and type 2 diabetes mellitus (T2DM). In fact, this association with abnormal glucose tolerance tests was reported as early as 1996 [10]. Numerous clinical studies reporting lower serum/plasma bilirubin concentrations with the risk of diabetes (and its complications) were published in the following decade, as summarized in 2012 [11], as well as being confirmed in more recent studies [12,13,14,15,16,17].

The opposite is also true, since subjects with mild unconjugated hyperbilirubinemia due to Gilbert’s syndrome are less likely to develop T2DM. Inoguchi et al. were the first to demonstrate that diabetic patients with concomitant Gilbert’s syndrome have a lower prevalence rate of vascular complications compared to normobilirubinemic diabetics [18].

In Caucasians, Gilbert´s syndrome is caused by the UGT1A1*28 polymorphism (the A(TA)_7_TAA variation of the promoter of the *UGT1A1* gene (OMIM*191740), rs8175347) that substantially reduces the glucuronosylation of bilirubin. Compared to normal cases, UGT1A1*28 homozygosity leads to a reduction in UGT1A1 activity by approximately 30% [19]. However, the penetrance of UGT1A1*28 homozygosity is incomplete, reaching only 50% [3]. On the other hand, in Asian populations, other mutations including heterozygous mutation in the coding exon 1 of the *UGT1A1* gene (such as a presence of the UGT1A1*6 allele) or other variants in the promoter *UGT1A1* region (such as mutations in the phenobarbital-responsive enhancer module NR3 region (gtPBREM NR3)) may predispose patients to hyperbilirubinemia [3]. Potent protective effects of mild hyperbilirubinemia on oxidative stress were repeatedly observed in Gilbert´s syndrome subjects, accounting for, at least in part, the clinical benefits of mildly elevated serum bilirubin concentrations [5,20,21].

Although based on large genome-wide association studies the *UGT1A1* gene has been found to be the major genetic determinant of serum bilirubin concentrations [22], its genetic variations have never been studied in association with the risk of T2DM.

In addition, heme oxygenase-1 (encoded by *HMOX1*, OMIM*141250) is another enzyme that contributes significantly to both bilirubin production and oxidative stress defense [22]. Microsatellite variations in the promoter regions of the *HMOX1* and *UGT1A1* genes have been implicated in the pathogenesis of oxidative stress-mediated diseases [23,24]. In fact, a higher number of (GT)n repetitions in the *HMOX1* gene promoter (categorized as class L allele) leads to underexpression of *HMOX1* [23] and are of possible clinical significance also in T2DM, as documented by a higher risk of T2DM [25] and a worse clinical outcome [26] in these patients, which were also associated with lower serum bilirubin concentrations [27].

Therefore, the aim of this study was to assess whether serum bilirubin concentrations and (TA)n microsatellite variations in the promoter regions of the *UGT1A1* and *HMOX1* genes are associated with T2DM in the Czech population.

## 2. Results

The basic clinical and laboratory parameters are stated in Table 1. The mean age of the patients with T2DM was 63.7 ± 10.8 years, with the mean duration of the disease being 10.8 ± 8.5 years. The patients were compensated on oral antidiabetic treatment and/or insulin, and a substantial proportion had micro- and macrovascular complications of diabetes (Table 1).

Compared to controls, patients with T2DM had significantly lower serum bilirubin concentrations (9.2 vs. 12.1 μmol/L, *p* < 0.001), both in male and female patients (9.9 vs. 12.9 μmol/L, *p* < 0.001, and 9.0 vs. 10.6 μmol/L, *p* < 0.001 in men and women, respectively, Table 2).

Each micromolar increase in serum bilirubin concentrations was associated with 11% (OR; 95% CI: 0.89; 0.85–0.92, *p* < 0.001), 12% (OR; 95% CI: 0.88; 0.83–0.93, *p* < 0.001), and 10% (OR; 95% CI: 0.90; 0.85–0.96, *p* < 0.001) decreases in the risk of T2DM development in the overall, male and female populations, respectively.

The prevalence of Gilbert´s syndrome (defined phenotypically based on serum bilirubin concentrations above 17.1 μmol/L in the absence of overt hemolysis and abnormal ALT activities) in patients with T2DM was significantly lower in the overall population (5.5 vs. 26%, *p* < 0.001), as well as in both men (5.7 vs. 30.2, *p* < 0.001) and women (5.3 vs. 20.6, *p* < 0.001), compared to our control population selected for their healthy status. The prevalence rate of Gilbert´s syndrome was also lower when compared to the Czech general population (8.9, 11.6 and 6.1% in the overall, male and female populations, respectively [28].

As expected, the presence of the (TA)_7_ allele was responsible for higher serum bilirubin concentrations in both T2DM patients and healthy controls, with the highest concentrations in UGT1A1 (TA)_7/7_ homozygotes (Table 3). However, compared to healthy controls, patients with T2DM had significantly lower serum bilirubin concentrations in each (TA)n genotype in both men and women (Table 3). The frequency of Gilbert´s syndrome genotype ((TA)_7/7_ homozygosity) was significantly higher in healthy subjects compared to T2DM patients, but this phenomenon was observed only in men and not women (Table 3).

Although a decreasing trend in serum bilirubin concentrations dependent on the presence of L allele was observed (especially in diabetic patients), *HMOX1* SL genotypes did not have a significant effect on serum bilirubin concentrations in healthy control subjects, either men or women (Table 4). Furthermore, the frequencies of individual *HMOX1* genotypes did not differ between patients with T2DM and healthy subjects. On the other hand, serum bilirubin concentrations were substantially lower in patients with T2DM compared to healthy controls in almost all genotypes (Table 4).

## 3. Discussion

Bilirubin has for the last few decades been recognized as a potent endogenous substance that counteracts the increased oxidative stress defense mechanism, affecting the immune system as well as intermediary metabolism. In fact, bilirubin behaves as a signaling molecule acting on various nuclear and cytoplasmic receptors [3,4]. Apart from beneficial effects on atherosclerotic diseases [29], mildly elevated serum bilirubin concentrations seem to protect against the development of obesity, metabolic syndrome [7,8,9] as well as diabetes [10,11,12,13,14,15,16,17,18].

In our present study in the Czech Caucasian population, we demonstrate a negative association of serum bilirubin concentrations with T2DM for both men and women, with a significant effect of serum bilirubin on the risk of T2DM. The prevalence of Gilbert´s syndrome among patients with T2DM was only 5.5%, which is much less compared to both the healthy Czech population and the general Czech population [28]. This observation is very similar to that of Inoguchi et al., who found that the occurrence of Gilbert´s syndrome was at least three times less likely among more than 5000 consecutive diabetics than would be expected in the general population [18]. Consistent with these data, a low prevalence of metabolic syndrome in subjects with phenotypic Gilbert´s syndrome was described in a recent large Korean study on more than 12,000 participants [30], and the same association was described in our recent study with NAFLD (non-alcoholic fatty liver disease) patients [31].

Interestingly, the frequency of the (TA)_7/7_
*UGT1A1* genotype was significantly low in the male, but not female, T2DM patients, indicating some contribution of genetic background in modifying T2DM manifestation. On the other hand, serum bilirubin concentrations were significantly lower in all (TA)_7/7_
*UGT1A1* genotypes (Table 3), most likely due to the consumption of bilirubin from increased oxidative stress accompanying T2DM. In this respect, it should also be noted that *UGT1A1* expression is under the control of estrogens, and this phenomenon accounts for the lower serum bilirubin concentrations, as well as the less pronounced protective effects of bilirubin, in women compared to men [32].

A similar analysis of (GT)n *HMOX1* genetic variations in our Czech patients did not reveal any impact on the risk of T2DM or any effect on serum bilirubin concentrations (Table 4). Our data do not confirm the results of recent a meta-analysis of six previously reported studies covering 1,751 diabetic patients, in which the presence of L allele in the promoter region of the *HMOX1* gene was associated with a small 12% increase in T2DM risk [25]. Simultaneously, our data do not confirm the association between (GT)n *HMOX1* polymorphism and serum bilirubin previously reported in a Taiwanese study [27].

The negative association of serum bilirubin with T2DM observed in our study as well as other previous studies seems to be plausible when taking into account the signaling activities of bilirubin reported in recent studies (as reviewed in [2,3,4]), in particular its beneficial effects on PPARα [33] and AMPK signaling [34]. As reviewed recently, bilirubin acting via these signaling pathways exerts real endocrine effects, which redefines bilirubin as a metabolic hormone that is essential for balancing the homeostasis of energetic sources [2,3,4]. Indeed, the key antidiabetic activity of bilirubin seems to be mediated via its PPARα-agonistic effects [33], which have been validated in various in vitro and in vivo studies (for review see [2]). In fact, studies on human hepatoblastoma HepG2 cells with PPARα knocked down demonstrated that the bilirubin-mediated expression of metabolic genes is mostly PPARα-dependent [35].

These observations are supported also by experimental data on hyperbilirubinemic Gunn rats who were found to resist the development of diabetes after intraperitoneal exposure to streptozocin compared to their normobilirubinemic littermates [36]. In fact, markers of diabetes, such as fasting blood glucose and HbA1 concentrations, were much less pronounced in Gunn rats with the preservation of insulin secretion by the pancreatic islets of these hyperbilirubinemic animals [36].

There are several possible limitations to our study, which was a retrospective, cross-sectional, non-randomized study. Hence, the possibility of selection bias cannot be fully excluded. For comparison with our patients with T2DM, we intentionally involved a cohort of control subjects selected for their healthy status, since general population subjects contain also subjects with various morbidities, including diabetes. Because of that, our healthy control subjects were substantially younger (39 ± 11 years). In fact, increasing age has been demonstrated to play a role in decreasing serum bilirubin concentrations in men (by 0.029 μmol/L with each year), but not in women in a large NHANES study [37]. Taking into account the difference between both cohorts in our study, the potential effect of age difference would represent a decrease in serum bilirubin concentrations in the healthy control men of less than 0.8 μmol/L.

Similarly, we did not obtain accurate information on the smoking status of our diabetic patients (while as many as 24.3% of our healthy cohort were smokers). Although it is well known that smoking is an important factor responsible for a decrease in the serum bilirubin, this decrease between smokers and non-smokers accounts for only a 7% decrease in serum bilirubin concentrations, as we proved in our previous study on the Czech general population [30]. Therefore, we conclude that neither smoking status nor age could substantially affect the large difference in serum bilirubin concentrations between T2DM patients and the general population, as observed in our study.

## 4. Materials and Methods

### 4.1. Patients

This study was performed on 220 patients with T2DM (Table 1) who had been consecutively examined in the 3rd Department of Internal Medicine, General University Hospital in Prague between 2007 and 2011. T2DM was diagnosed according to the definition of diabetes of the American Diabetes Association. To eliminate a possible confounding effect of underlying liver disease, only subjects with physiological values of ALT activities (<0.78 μkat/L) were included, meaning that the patients did not have any underlying liver disease. At the same time, according to laboratory data, clinically relevant hemolytic disease was also excluded in all patients.

The control group consisted of 231 sex-matched control subjects who had been selected for their health status based on the absence of any acute or chronic disease, representing a general population sample from the same geographical region. These subjects were recruited from healthy blood donors and employees of the General University Hospital. All subjects in both cohorts were of Caucasian ancestry [38].

Written informed consent was obtained from each human subject included in the study, and the study protocol was consistent with the ethical guidelines of the Declaration of Helsinki of 1975, as reflected in the a priori approval by the Ethics Committee of the General University Hospital in Prague.

### 4.2. Laboratory Methods

In all subjects, the standard serum biochemistry was determined by routine assays on an automated analyzer (Cobas R8000 Modular analyzer, Roche Diagnostics GmbH, Mannheim, Germany).

### 4.3. DNA Analysis

Genomic DNA was isolated from peripheral white blood cells by means of a standard salting out method. The (GT)n variations in the *HMOX1* (dbSNP rs1805173), and (TA)n variations in the *UGT1A1* (dbSNP rs81753472) gene promoters were simultaneously determined using multicolored capillary electrophoresis as previously described [39]. The corresponding DNA fragments were amplified by polymerase chain reaction (PCR) using the following primers: (for *HMOX1*: forward 5′-CTGCAGCTTCTCAGATTTCC-3′; reverse 5′-ACAAAGTCTGGCCATAGGAC; for *UGT1A1*: forward 5′-GAACTTGGTGTATCGATTGGTTTTGC-3′; reverse 5′-CATCCACTGGGATCAACAGTATCTTCC-3′). The reverse primers were labeled at the 5′ end with WellRED fluorescent dyes (Beckman Coulter, Fullerton, CA, USA). PCR products were separated on a CEQ 8000 Genetic Analysis System (Beckman Coulter, Fullerton, CA, USA). The length variations of the *HMOX1* (GT)n repeats were classified into short S (n < 27) and long L (n ≥ 33) subgroups.

### 4.4. Statistical Analyses

Data are expressed as mean ± SD, or median and IQ range when the data were not normally distributed. Cohort data were compared using Student’s *t* test or the Mann–Whitney rank sum test, depending on data normality. The percentage counts were compared using the chi-square test. ANOVA on ranks with Dunn’s post hoc testing was used to compare variables among individual groups of patients. Logistic regression analysis was used to assess the effect of serum bilirubin concentrations on the risk of T2DM.

All analyses were performed with the alpha set to 0.05. Statistics were calculated using SigmaPlot v. 14.5 (Systat Software Inc., San Jose, CA, USA).

## 5. Conclusions

In conclusion, our results demonstrate that the manifestation of T2DM is associated with low serum bilirubin concentrations. Consumption of bilirubin due to the increased oxidative stress associated with T2DM appears to be the main explanation for this, although (TA)n repeat variations in *UGT1A1* partially contribute to this phenomenon. Hence, nutraceutical/pharmacotherapeutic approaches focused on targeting the heme catabolic pathway [17,40] seem promising in the chemoprevention or treatment of T2DM.

## Figures and Tables

**Table 1 ijms-24-10614-t001:** Basic clinical and laboratory parameters of T2DM patients and control subjects.

Parameter	T2DM Patients (n = 220)	Controls (n = 231)	*p*-Value
**Age** (y)	63.7 ± 10.8	38.6 ± 10.9	<0.05
**M:F ratio**	0.93	1.26	NS
**Disease duration** (y)	10.8 ± 8.5	-	NA
**Fasting glucose** (mmol/L)	8.9 ± 3.6	4.7 ± 0.6	<0.05
**HbA1** (mmol/mol)	46 ± 14.5	ND	NA
**ALT** (μkat/L)	0.42 ± 0.2	0.40 ± 0.2	NS
**Therapy with OAD** (%)	39.7	NA	NA
**Therapy with insulin** (%)	26.7	NA	NA
**Therapy with insulin plus OAD** (%)	33.6	NA	NA
**Systolic blood pressure** (mm Hg)	137.4 ± 18	118.3 ± 16	<0.05
**Diastolic blood pressure** (mm Hg)	77.5 ± 10.8	78.7 ± 9	NS
**Retinopathy and/or nephropathy** (%)	27.6	NA	NA
**Atheroclerotic diseases** (%)	38.6	NA	NA

Data expressed as mean ± SD or %. OAD, oral antidiabetics; NA, not applicable; ND, not done; NS, not significant, atheroclerotic diseases include coronary heart disease, chronic limb ischemia and/or stroke.

**Table 2 ijms-24-10614-t002:** Serum bilirubin concentrations in T2DM patients.

	Serum Bilirubin [μmol/L]		
Group	T2DM Patients (n = 220; M = 106; F = 114)	Controls (n = 231; M = 129; F = 102)	*p*-Value
**Total**	**9.2** [7.1–12.2]	**12.1** [9.0–17.5]	<0.001
**Males**	**9.9** [7.3–12.7]	**12.9** [10.1–18.5]	<0.001
**Females**	**9.0** [6.9–12.1]	**10.6** [8.4–15.5]	<0.001

Data expressed as median and IQ range.

**Table 3 ijms-24-10614-t003:** Serum bilirubin concentrations in diabetic patients according to *UGT1A1* (TA)_7_ status.

*UGT1A1* (TA)_n_	T2DM Patients		Controls		*p*-Value
	**All**				
	**n** (%) *	**Bilirubin** (μmol/L)	**n** (%) *	**Bilirubin** (μmol/L)	
**(** **TA)_6/6_**	**43.2**	**8.2** [6.4–11.0]	**31.8**	**10.0** [7.7–12.2]	**0.013**
**(TA)_6/7_**	**44.7**	**9.9** [7.4–12.4]	**43**	**13.0** [9.4–16.3]	**<** **0.001**
**(** **TA)_7/7_**	**12.1**	**11.3** [9.2–16.7]	**25.2**	**19.3** [11.8–29.6]	**<** **0.001**
***p*** **for trend**		**<** **0.001**		**<** **0.001**	
	**Men**				
	**n** (%) **	**Bilirubin** (μmol/L)	**n** (%) **	**Bilirubin** (μmol/L)	
**(** **TA)_6/6_**	**48.5**	**8.5** [6.4–11.0]	**29.5**	**10.3** [9.0–12.2]	**0.025**
**(TA)_6/7_**	**42.6**	**10.9** [7.5–14.9]	**38.7**	**14.3** [10.6–18.2]	**0.002**
**(** **TA)_7/7_**	**8.9**	**11.0** [9.9–14.6]	**31.8**	**17.7** [11.4–30.5]	**0.019**
***p*** **for trend**		**0.005**		**<** **0.001**	
	**Women**				
	**n** (%) ***	**Bilirubin** (μmol/L)	**n** (%) ***	**Bilirubin** (μmol/L)	
**(** **TA)_6/6_**	**38.1**	**7.8** [6.3–11.1]	**35.3**	**9.5** [6.9–12.1]	**NS**
**(TA)_6/7_**	**46.7**	**9.1** [7.3–11.5]	**49.4**	**10.6** [8.7–14.4]	**0.008**
**(** **TA)_7/7_**	**15.2**	**11.8** [7.7–16.9]	**15.3**	**20.8** [17.0–26.3]	**0.008**
***p*** **for trend**		**0.012**		**<** **0.001**	

Serum bilirubin expressed in μmol/L, data given as median and IQ range. * *p* = 0.001; ** *p* < 0.001, *** NS, comparison of individual genotype frequencies between T2DM patients and healthy controls (based on the chi-square test). NS, not significant. ANOVA on ranks with Dunn’s post hoc testing was used to compare *p* for trends of serum bilirubin concentrations among individual genotypes.

**Table 4 ijms-24-10614-t004:** Serum bilirubin concentrations in diabetic patients according to *HMOX1* SL status.

*HMOX1* (GT)_n_	T2DM Patients		Controls		*p*-Value
	**All**				
	**n** (%) *	**Bilirubin** (μmol/L)	**n** (%) *	**Bilirubin** (μmol/L)	
**SS**	**15.0**	**13.0** [9.5–17.5]	**14.9**	**11.3** [7.4–13.4]	**NS**
**SL**	**41.1**	**8.8** [6.7–11.0]	**43.9**	**12.7** [9.0–17.6]	**<** **0.001**
**LL**	**43.9**	**9.1** [7.1–12.3]	**41.2**	**12.0** [9.1–17.6]	**<** **0.001**
***p*** **for trend**		**NS**		**NS**	
	**Men**				
	**n** (%) *	**Bilirubin** (μmol/L)	**n** (%) *	**Bilirubin** (μmol/L)	
**SS**	**11.7**	**10.2** [7.0–12.1]	**13.6**	**12.1** [9.1–14.4]	**NS**
**SL**	**41.7**	**10.1** [6.8–13.3]	**44.6**	**12.9** [9.2–19.4]	**<** **0.001**
**LL**	**46.6**	**9.2** [7.4–13.0]	**41.8**	**13.6** [10.6–20.4]	**<** **0.001**
***p*** **for trend**		**NS**		**NS**	
	**Women**				
	n (%) *	**Bilirubin** (μmol/L)	**n** (%) *	**Bilirubin** (μmol/L)	
**SS**	**18.0**	**12.2** [7.6–14.3]	**16.7**	**14.2** [10.3–18.5]	**NS**
**SL**	**40.5**	**8.2** [6.5–10.1]	**42.8**	**11.4** [8.6–15.9]	**0.001**
**LL**	**41.4**	**9.1** [6.7–11.6]	**40.5**	**10.1** [7.9–13.5]	**NS**
***p*** **for trend**		**NS**		**NS**	

Serum bilirubin expressed in μmol/L, data given as median and IQ range. The length variations of *HMOX1* (GT)n repeats were classified into the short (S, n < 27) and long (L, n ≥ 33) allele subgroups. * *p* = NS, comparison of frequencies of individual genotypes among T2DM patients and healthy controls (based on chi-square test). NS, not significant. ANOVA on ranks with Dunn’s post hoc testing was used to compare *p* for trend of serum bilirubin concentrations among individual genotypes.

## Data Availability

All research data are available on request from the corresponding author.

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
