# Peer review of "Association of Low Serum Bilirubin Concentrations and Promoter Variations in the UGT1A1 and HMOX1 Genes with Type 2 Diabetes Mellitus in the Czech Population"

_ijms, 2023, doi:10.3390/ijms241310614_

Round 1
Reviewer 1 Report
As laboratory medicine specialist I would suggest adding the level of hemoglobin to the analysis. Bilirubin is clearly a metabolite, its low level may express low hemoglobin – perhaps should be taken into consideration...
Author Response
Point-by-point reply to Reviewers´ comments
We would like to thank the Editor and reviewers for their evaluation as well as their comments. We have tried to address all these issues and believed that our Ms. has been now been improved substantially. All changes in the Ms. have been highlighted with yellow color.
Reviewer 1
Comment: As laboratory medicine specialist I would suggest adding the level of hemoglobin to the analysis. Bilirubin is clearly a metabolite, its low level may express low hemoglobin – perhaps should be taken into consideration...
Reply: Although we fully agree that the hemoglobin concentration is a possible predictor of serum bilirubin concentrations, the opposite relationship is true, i.e. subjects with increased red blood cell breakdown (hemolysis) have lower hemoglobin levels and simultaneously increased serum bilirubin concentrations.
Unfortunately, these data are already not available to the authors. Pls note, that the study is retrospective, based on datasets retrieved between 2007-2011. Nevertheless, all subjects had at the time of enrollment complete clinical and laboratory examinations which did not reveal any pathological signs of increased hemolysis. We acknowledged this fact in the MM Section (see p. 7).
Reviewer 2 Report
Introduction - paragraph about Gilbert sy. - briefly mention the role of enhancer mutation
Methods - study sample selection - please describe in more details (e.g. consecutive patients from diabetology clinic, address the possibility of selection bias)
Methods - control sample - age and sex matched?
Results - please include baseline comparisons for age, sex, BMI, ALT between study and control groups
Discussion-
Discrepancy in GS prevalence in healthy controls and general czech population may affect the difference in observed levels of bilirubin and thus the significance of the difference between healthy controls and T2DM patients
Author Response
Point-by-point reply to Reviewers´ comments
We would like to thank the Editor and reviewers for their evaluation as well as their comments. We have tried to address all these issues and believed that our Ms. has been now been improved substantially. All changes in the Ms. have been highlighted with yellow color.
Reviewer 2
Comment 1:
Introduction - paragraph about Gilbert sy. - briefly mention the role of enhancer mutation
Reply: As recommended, information about the role of phenobarbital-responsive enhancer module mutation of UGT1A1 gene promoter has been added into the Introduction Section (see page 2).
Comment 2:
Methods - study sample selection - please describe in more details (e.g. consecutive patients from diabetology clinic, address the possibility of selection bias)
Reply: More details about study populations have been added in the MM Section, and comment on possible selection bias has been added into Limitations of the study in the Discusion Section (see p… 2 and 6-7).
Comment 3:
Methods - control sample - age and sex matched?
Reply: The control subjects were sex-matched (this info has been added in the MM Section, see p. 7), but not age-matched. As we explain in detail in the Discussion Section (see p. 6-7), we have intentionally control subjects selected for healthy status not to include subjects with any comorbidities which could have confound our comparisons. Because age may affect serum bilirubin concentrations, we have discussed this effect and its possible role on our data in the Discussion Section (see p. 7).
Comment 4:
Results - please include baseline comparisons for age, sex, BMI, ALT between study and control groups.
Reply: Baseline comparisons of basic variables between studied populations have been added in the Table 1.
Comment 5:
Discussion- Discrepancy in GS prevalence in healthy controls and general Czech population may affect the difference in observed levels of bilirubin and thus the significance of the difference between healthy controls and T2DM patients.
Reply: We are very well aware of this fact. The major implication of our observations is that when compared with healthy population, diabetics had substantially deteriorated bilirubin metabolism. This effect is obviously less pronounced when compared to general population, which contain also subjects various morbidities, including diabetes. We have better explained these facts in the Discussion Section (see p. 6-7).
Reviewer 3 Report
In the article " Association of low serum bilirubin concentrations and promotore variations in the UGT1A1 and HMOX1 genes with type 2 diabetes mellitus in the Czech population" is described the results from a retrospective cohort of patients versus controls about the levels of serum bilirubin and the polymorphisms in the promoter of these to genes, establishing an association between them in T2DM patients, considering low levels of biliribun and a decrease in the repetitions on the promoter of UGT1A1 gene, and less incidence of Gilbert syndrome, but this association is not relevant for gene HMOX1.
From my point of view, the article is very descriptive and associative, not deeping in the molecular insights of those findings. The authors could improve the article by including new data from the patients and also the controls, that could reinforce their conclusions, as analysing the expression of the gene (which is the impact of the polymorphisms in the promoter in the gene expression?), or why this causal relationship between more indicence of Gilbert syndrome --> more bilirubin --> more protection --> less diabetes, which is the redox status of the patients (GSH ratio, 8-oxoguanidine status?). Furthermore, they found differences between men and women, but not a plausible explanation is given.
On the other hand, regarding the data, the descriptive results from control cohort are missing in Table 1, and data presented in the Table 3 is not necessary to show in a table format., as well as table 4 that could be included in table 2 and within the text.
Besides that, text should be reviewed in general more accurately, as there are some explanations of the template remaining in the text (lines 221-222), as an example.
Author Response
Point-by-point reply to Reviewers´ comments
We would like to thank the Editor and reviewers for their evaluation as well as their comments. We have tried to address all these issues and believed that our Ms. has been now been improved substantially. All changes in the Ms. have been highlighted with yellow color.
Reviewer 3
Comment 1:
From my point of view, the article is very descriptive and associative, not deeping in the molecular insights of those findings. The authors could improve the article by including new data from the patients and also the controls, that could reinforce their conclusions, as analysing the expression of the gene (which is the impact of the polymorphisms in the promoter in the gene expression?), or why this causal relationship between more incidence of Gilbert syndrome --> more bilirubin --> more protection --> less diabetes, which is the redox status of the patients (GSH ratio, 8-oxoguanidine status?). Furthermore, they found differences between men and women, but not a plausible explanation is given.
Reply: We thank the reviewer for this important comment. However, suggested analyses were either clinically difficult or impossible to performed (gene expression studies) or beyond the scope of this study (studies on redox status) since these have already been described in the literature (including our own studies). We have properly discussed all suggested issues, including differences between men and women as follows:
a) UGT1A1 expression analyses: UGT1A1*28 homozygosity, responsible for the manifestation of Gilbert´s syndrome in most Caucasians subjects, leads to approximately 30% activity of UGT1A1 compared to normals (Bosma. The genetic basis of the reduced expression of bilirubin UDP-glucuronosyltransferase 1 in Gilbert's syndrome. New Engl J Med 1995.). This information was added in the Introduction Section (see p. 1-2).
b) Redox status analyses: Potent protective effects of mild hyperbilirubinemia on oxidative stress were repeatedly observed in Gilbert syndrome subjects accounting, at least in part, for clinical benefits of mildly elevated serum bilirubin concentrations (Vitek et al. Gilbert syndrome and ischemic heart disease: a protective effect of elevated bilirubin levels. Atherosclerosis 2002; Vitek et al. Relationship between serum bilirubin and uric acid to oxidative stress markers in Italian and Czech populations. J Appl Biomed 2013; Wagner et al. Oxidative stress and related biomarkers in Gilbert's syndrome: A secondary analysis of two case-control studies. Antioxidants 2021). This information was added in the Introduction Section (see p. 2).
c) Explanation on sexual dimorphism: Differences between men a women in bilirubin metabolism observed in our and also previous studies is due to the fact that UGT1A1 expression is under the control of estrogens (Wagner et al. Looking to the horizon: the role of bilirubin in the development and prevent on of age-related chronic diseases. Clin Sci 2015). This information was added in the Discussion Section (see p. 6).
Comment 2:
On the other hand, regarding the data, the descriptive results from control cohort are missing in Table 1, and data presented in the Table 3 is not necessary to show in a table format., as well as table 4 that could be included in Table 2 and within the text.
Reply: Control cohort data have been added into Table 1, and data in Tables 3 and 4 have been adjusted as recommended (see page 3).
Comment 3:
Besides that, text should be reviewed in general more accurately, as there are some explanations of the template remaining in the text (lines 221-222), as an example.
Reply: Text was carefully reviewed and all the inappropriate content was removed.
Round 2
Reviewer 3 Report
Thanks for the clarifications, those improved the text, although it is mainly descriptive. I understand the limitations of the cohort, and the availability of the samples, but could be possible to provide more molecular insights from in silico analysis from human datasets or some other experiments in animal models. The quality is appropriate, but I consider the contents could have more soundness with additional data as suggested.
Author Response
We thank the Reviewer for the additional comment. We have added explanatory paragraph (p. 6, highlighted in yellow color) on the role of bilirubin-PPARa interaction in modulation of glucose metabolism and T2DM pathogenesis.
Round 3
Reviewer 3 Report
The article is improved from the first version, and the paragraph added in the conclusion introduces some insights about the molecular mechanisms involved.
Author Response
We would like to thank the reviewer for his/her positive comments.